# PROTEIN LANGUAGE MODELS ENABLE ACCURATE CRYPTIC LIGAND BINDING POCKET PREDICTION

## ABSTRACT

Accurate prediction of protein-ligand binding pockets is a critical task in protein functional analysis and small molecule pharmaceutical design. However, the flexible and dynamic nature of proteins conceal an unknown number of potentially invaluable "cryptic" pockets. Current approaches for cryptic pocket discovery rely on molecular dynamics (MD), leading to poor scalability and bias. Even recent ML-based cryptic pocket discovery approaches require large, post-processed MD datasets to train their models. In contrast, this work presents "Efficient Sequence-based cryptic Pocket prediction" (ESP) leveraging advanced Protein Language Models (PLMs), and demonstrates significant improvement in predictive efficacy compared to ML-based cryptic pocket prediction SOTA (ROCAUC 0.93 vs 0.87). ESP achieves detection of cryptic pockets via training on readily available, non-cryptic-pocket-specific data from the PDBBind dataset, rather than costly simulation and post-processing. Further, while SOTA's predictions often include positive signal broadly distributed over a target structure, ESP produces more spatially-focused predictions which increase downstream utility.

## 1 INTRODUCTION

The Transformer architecture (Vaswani et al., 2023) arose in the context of machine translation, and also enables SOTA applications in text-classification (Devlin et al., 2019) and text-generation (Radford et al., 2018). Large Language Models (LLMs) scale the Transformer using more blocks, larger embeddings, and larger datasets to achieve unprecedented performance on a variety of tasks in the zero-shot setting and now exhibit sophisticated knowledge of semantic relationships (Brown et al., 2020).

A core hypothesis of structural biology is that structure and function arise from specific sequences of amino acids, in the same way that meaning in natural langauge arises from specific sequences of words. Since Transformers are agnostic to the meaning of semantic tokens, so-called Protein Language Models (PLMs) were trained using the same Masked Language Model pre-training objective as Devlin et al. (2019). Rives et al. (2019) found that PLMs can not only learn key differences between amino acids themselves, but also distinguish which proteins have similar structures yet different sequences in the zero-shot setting. Rao et al. (2020) showed that PLMs learn what structural biologists call contact maps in their self-attention coefficients, with a linear relation between preplexity and "contact precision." Lin et al. (2022) found that scaling a PLM improves performance on the aforementioned, and enables structure prediction competitive with AlphaFold2 (Jumper et al., 2021). These results are an astonishing confirmation of the protein sequence-structure-function hypothesis, and motivate investigating the fronteirs of what PLMs can power.

For example, Singh et al. (2023) used PLMs and molecular fingerprints to conduct virtual screening that successfully identified sub-nanomolar binders. The method was simple: (1) a protein's PLM `[CLS]` embedding and ligand's molecular fingerprint (Glem et al., 2006) were projected into the same dimensional space using a single linear layer followed by ReLU, (2) cosine similarity was calculated, and (3) the projectors were updated via contrastive learning. While this application is interesting, the real significance is that information sufficient for this purpose could be embedded into a single summary token by a PLM.

Small molecule pharmaceutical discovery often leverages insight from structural biology data, and PLMs offer a new window into this domain. Specifically, knowledge of where compounds, or "lig-

ands," bind to a protein of interest is of critical importance, providing a starting point for medicinal chemists to design better molecules. This work focuses on the use of PLMs to identify hard-to-find, or "cryptic," protein pockets from sequence alone, and in particular the following three specific aims: (1) explore the relevance of SOTA PLMs for sequence-based cryptic protein-ligand binding pocket prediction, (2) determine the extent to which multi-task learning with secondary structure prediction (SSP) enhances cryptic pocket prediction, and (3) offer a specific model that redefines SOTA for cryptic protein pocket prediction. Toward these ends we find that: (1) many PLMs enable predictive efficacy beyond previous SOTA cryptic pocket prediction algorithms, and Ankh-Large (Elnaggar et al., 2023) and ESM-2 15B enable top AUC and APS, respectively, (2) multi-task learning with SSP enhances predictive efficacy for many cases, and (3) our ESP model outperforms the SOTA ML-based cryptic pocket prediction algorithm, PocketMiner (PM) (Meller et al., 2023), by a significant margin (ROCAUC 0.93 vs 0.87) on its own test set.

## 2 BACKGROUND

### 2.1 IMPACT OF CRYPTIC POCKET PREDICTION TO SMALL MOLECULE DRUG DESIGN

Cryptic protein-ligand binding pocket prediction is a high impact task because successful predictions can form the basis for novel structure-based small molecule pharmaceutical development programs. "Cryptic" or "non-obvious" pockets are so named because of the difficultly in recognizing such ligandable pockets with conventional tools. Whereas rigid, highly-conserved active sites and other non-cryptic pockets can commonly be identified by a structural biologist in receptor or enzyme protein structures determined via x-ray crystallography, cryptic pockets generally cannot. Instead, they are often discovered accidentally in experimentally solved protein structures in complex with wet-lab screening or fragment screening hits, or through extensive molecular dynamics (MD) simulations using existing structures.

Finding a cryptic pocket on a pharmacologically validated protein target can motivate resource allocation to produce a novel, first-in-class medication. New binding sites for validated targets enable development of new chemistry with potential for improvement in efficacy, dosing regimen, and reduction of side-effects. Significant improvement in one or a combination of those three clinical properties can improve the standard of care for patients.

Cryptic pocket prediction, and ligand binding pocket prediction in general, also has impact when engaging new targets for the first time. Because new targets may not have any reference compounds known to engage it, virtual screening and de novo molecular generation against a putative pocket may be required to find hits that medicinal chemists can turn into leads and eventually drug candidates.

### 2.2 NON-CRYPTIC POCKET IDENTIFICATION ALGORITHMS

Computational identification of ligand binding sites on protein surfaces is a field with many mature tools, each with their own capabilities and limitations. They evaluate structures at the atomic level, the residue level, or arbitrary grid points in space. These methods perform best when there is a cavity on the protein surface that looks similar in volume and chemistry to common, known binding sites. They tend not to elucidate cryptic pockets because they only see snapshots of geometric and chemical properties. Dynamic properties of proteins that might correlate with or imply a propensity for cryptic pockets to form are not taken into account.

Two notable examples of methods of this type are LIGSITE (Hendlich et al., 1997) and fpocket (Le Guilloux et al., 2009). LIGSITE scans a protein for concave volumes and reports cavities above a minimum size. Fpocket uses Voronoi tessellation to define alpha spheres which are then clustered prior to ranking clusters and scoring pockets. While dated by ML standards, these two algorithms are still relevant in compuational chemistry and were used during label generation by the SOTA ML cryptic pocket algorithm discussed below. For a review of non-ML and ML pocket prediction approaches developed over the past three decades see Zhao et al. (2020) and Di Palma et al. (2023).

## 2.3 ML-Based, Cryptic Pocket Identification Algorithms

Cryptic pocket identification algorithms aim to find hard-to-find areas of a protein where a drug can bind and achieve a disease-modifying effect. Simple geometric calculations to find concave surfaces will not suffice because cryptic pockets in experimentally-solved structures are generally not in a shape that allows a drug to bind. Cryptic pockets tend to form by protein atom movement opening a pocket, or bringing distal parts close enough to form a pocket. Since the flexibility of a protein enables these phenomena, MD is a tool for finding cryptic pockets. Unfortunately, scaling MD for this purpose is time consuming and cost prohibitive.

The SOTA ML approach to cryptic pocket prediction is PocketMiner (PM). It uses a protein's backbone atom coordinates and sequence to predict whether or not each residue is associated with the formation of a cryptic pocket. It does not however use information from SOTA PLMs. Its use case is to feed in protein structures absent any bound ligand, and then predict where cryptic pockets are most likely to form. PM was trained using labels generated by LIGSITE, fpocket, and conditional characterization of MD trajectories. It achieves improved accuracy and orders-of-magnitude improvement in inference compute cost compared to its predecessor CryptoSite (Cimermancic et al., 2016), which works best when MD simulations are executed at inference time. The small number of structures used to generate training data were enough to produce meaningful predictions on structures that are completely different from the training structures. Also, it achieves this with only 736,155 trainable parameters and no PLM.

## 3 Result

This work presents three major results: (1) training using samples with less than 30% sequence similarity to validation or test samples, (2) training across all levels of sequence similarity, and (3) projection of predictions onto structures and comparison with SOTA.

### 3.1 Performance at the 30% Sequence Identity Limit

For the first set of results, training samples having greater than 30% sequence identity with any validation or testing sample have been removed. This reduces data leakage in the structure domain, because samples with relatively low sequence identity can still be structurally similar. Amino acids can sometimes be changed without altering the overall structure or function of the protein, for example, if the amino acids are very similar or solvent exposed. While structural similarity can occur even when sequence identity is below 30%, results at the 30% threshold are still a meaningful measure of generalization ability. Sensitivity to this threshold will be addressed in the next section. Further detail is provided in the Methods section.

Figure 1 shows the best prediction head and task regime for the model with the highest APS and AUC for each PLM. Ankh-Large enables the best AUC of 0.926, and the best AUC for all prediction head classes except MLP. ESM-2 15B enables the best APS of 0.865, and the best MLP in both APS and AUC. ESM-2 15B also enables the best APS for MHA without `[CLS]` tokens. MHA using `[CLS]` tokens is the prediction head most common in Figure 1a, and PDBBind-label-only the most common training regime. MLP was the top ESM-2 15B prediction head in terms of AUC, and for ProtT5-XL on both APS and AUC. Multitask training using SSP resulted in top models for half of top MLPs, and was less common for either MHA architecture.

SOTA, PocketMiner, achieves 0.81 and 0.87 APS and AUC on the test set, respectively. Ankh-Large, ESM22 15B, ESM-2 3B, and ProtT5-XL all produced prediction heads of each class (MLP, MHA with `[CLS]`, and MHA without `[CLS]`) that outperformed SOTA in either or both of APS and AUC. No prediction head atop ProtBert outperformed SOTA on either APS or AUC.

Ankh-Large has an embedding size of 1536, compared to 5120 and 2560 for ESM-2 15B and ESM-2 3B, respectively. Both ProtT5-XL and ProtBert have an embedding size of 1024. The benefit of the higher APS achievable via ESM-2 15B is offset by significantly higher cost in terms of calculating the embeddings and training the prediction head. When a high AUC predictor is more appropriate, then Ankh-Large vastly outperforms both ESM-2 variants when computational cost is taken into account. If computational cost is the highest prority, users may wish to use ProtT5-XL embeddings with MLP prediction heads.

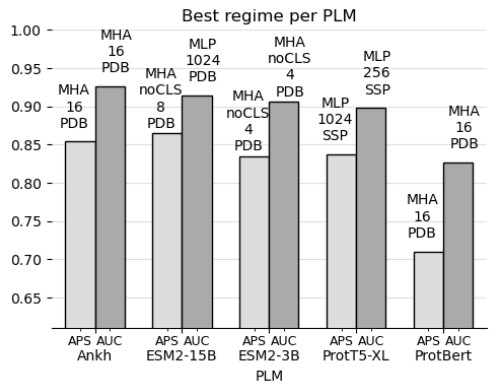
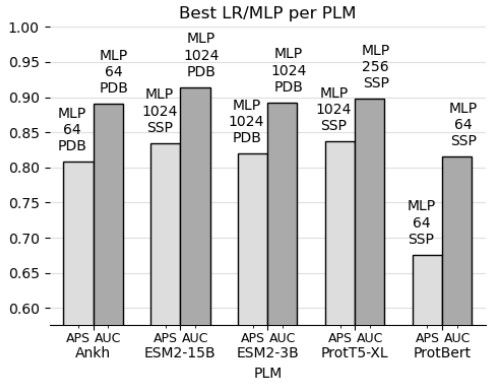

(a) Best regime per PLM and metric.

(b) Best MLP per PLM and metric.

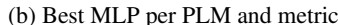

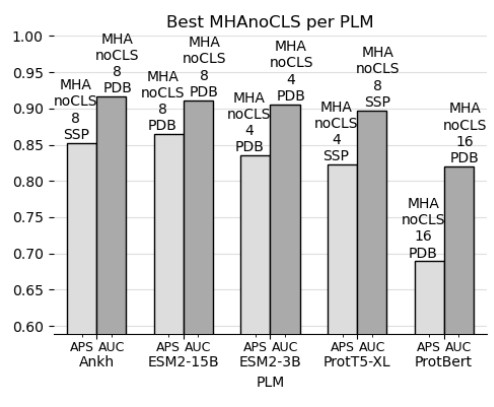
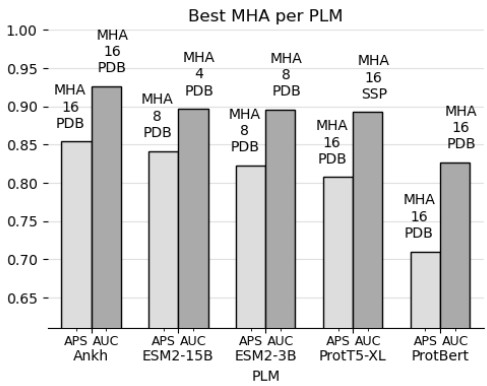

(c) Best MHA without CLS per PLM and metric.

(d) Best MHA per PLM and metric.

Figure 1: Figures 1 (a)-(d) show APS and AUC for the best prediction head for each PLM overall, and for each prediction head class. Text above each bar indicates the best in class prediction head. The integers indicate width of MLP or attention heads of MHA. PDB or SSP indicate the single- or multi-task setting, respectively.

### 3.1.1 ANKH-LARGE AT THE 30% SEQUENCE IDENTITY LIMIT

Performance metrics for ESP using Ankh-Large are summarized in Table 1. Table elements in bold indicate the best performance on either APC or AUC within a class of prediction heads. The prediction heads classes are: LR/MLP, MHA without `[CLS]` tokens, MHA using `[CLS]` tokens.

Ankh-Large achieves the best performance of all the PLMs under test in terms of AUC, and second best in terms of APS. All Ankh-Large prediction heads outperform SOTA for AUC, and many outperform SOTA for APS. Of the Ankh-Large MHA results, the best came from those trained on the single task of predicting the PDBBind labels, and MHAs using 4 attention heads tended to underperform. Multi-task training with SSP achieved the best APS for MHA without `[CLS]` tokens.

LR achieved APS and AUC of 0.805 and 0.889, respectively. This too outperforms SOTA on AUC. The high performance of LR suggests that the embeddings produced by Ankh-Large have a meaningful degree of linear correlation to the PDBBind-ligand-derived labels. Single layer MLPs of various widths struggled to make meaningful improvement beyond this baseline, with the best performing MLP having only 64 nodes (apparent using more significant figures).

Table 1: APS and ROCAUC results across architectures for ESP using Anhk-Large embeddings. Training samples with greater than 30% sequence identity with any member of validation or testing sets has been removed.

| PLM: Ankh-Large | PDBBind only | | w/ SSP | |
|---|---|---|---|---|
| Architecture | APS | AUC | APS | AUC |
| PocketMiner (PM) | 0.81 | 0.87 | N/A | N/A |
| LR | 0.805 | 0.889 | 0.805 | 0.889 |
| MLP 16 | 0.805 | 0.890 | 0.801 | 0.890 |
| MLP 64 | **0.808** | **0.890** | 0.807 | 0.890 |
| MLP 256 | 0.806 | 0.890 | 0.807 | 0.890 |
| MLP 1024 | 0.808 | 0.890 | 0.804 | 0.889 |
| MHA 4 no CLS | 0.845 | 0.911 | 0.755 | 0.891 |
| MHA 8 no CLS | 0.852 | **0.916** | **0.853** | 0.911 |
| MHA 16 no CLS | 0.820 | 0.897 | 0.841 | 0.908 |
| MHA 4 | 0.802 | 0.906 | 0.832 | 0.907 |
| MHA 8 | 0.821 | 0.908 | 0.849 | 0.897 |
| MHA 16 | **0.854** | **0.926** | 0.840 | 0.902 |

## 3.2 PERFORMANCE AS A FUNCTION OF SEQUENCE IDENTITY LIMIT

Figure 2 shows the best prediction head and task regime for the model with the highest APS and AUC for each sequence identity threshold. Ankh-Large was used for all models in this subsection.

Figure 2a shows that 7 of 12 best models were MHA using `[CLS]` tokens, whereas the remainder were MHA without use of `[CLS]` tokens. 4 of 12 best models were trained in the multi-task setting using both PDBBind ligand-derived labels and SSP labels. The best performing models were fairly consistent in terms of AUC until the 100% sequence identity threshold has used, when AUC rose to above 0.95. APS was less consistent, but trended upward overall with increased sequence identity threshold, achieving an APS of nearly 0.90 at the sequence identity threshold of 100%.

Figure 2b shows performance as a function of sequence identity limit for LR/MLP prediction heads. The upward trending performance as sequence identity threshold is increase is smooth yet slight for AUC, and again more varied but overall uptrending for APS.

The relatively flat performance curve until the 100% sequence identity level is consistent with expectations of a well generalized model. One potential confounder arises when multiple structures with different labels for the same residues enter the training set as the sequence identity limit increases. Detection, analysis, and mitigation of this possibility is left for future work.

## 3.3 INFERENCE ON TEST SET EXAMPLES

The PM test set offers two types of samples. The first has only positive labels and "unknown" labels, and the second has only negative labels and "unknown" labels. Positive labels indicate residues known to be associated with cryptic pocket formation, whereas negative labels indicate residues known to not be associated with cryptic pocket formation. "Unknown" labels indicate residues where cryptic pocket formation is neither known to occor nor known to not occur, and are masked when calculating APS and AUC.

### 3.3.1 E. COLI OUTER MEMBRANE TRANSPORTER FECA (1KMO)

Figure 3 shows inference results and labels for E. coli outer membrane transporter FecA (RCSB PDB ID 1KMO). The ESP inference results for 1KMO, Figure 3a, show positive signal focused in the area of the cryptic pocket positive labels shown in Figure 3c, which are the PM test set labels for this protein. Outside the area of the cryptic pocket labels, ESP predicts that no cryptic pockets are present. This prediction is easy to interpret, and phenomenologically correct in the sense that the known pocket area stands out as such.

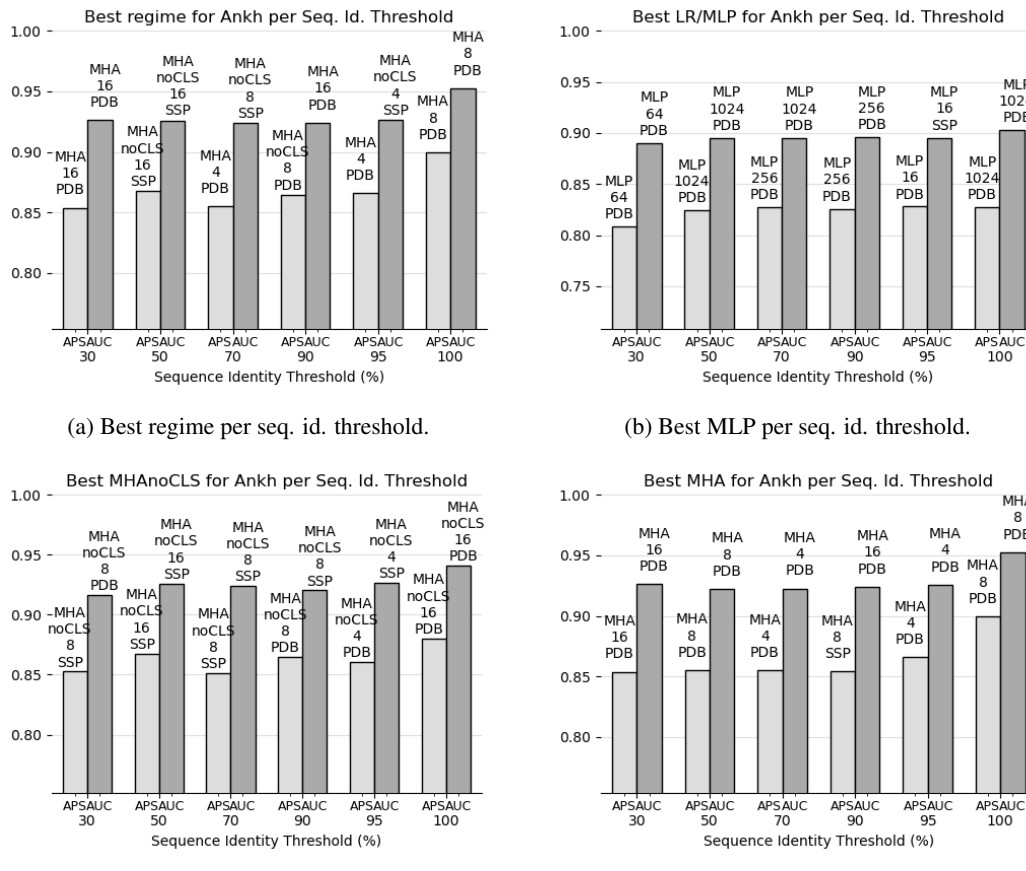

(a) Best regime per seq. id. threshold.

(b) Best MLP per seq. id. threshold.

(c) Best MHA without CLS per seq. id. threshold.

(d) Best MHA per seq. id. threshold.

Figure 2: Figures 2 (a)-(d) show APS and AUC for the best prediction head atop Ankh-Large overall, and for each prediction head class. (Text above each bar as above.)

The PM inference results for 1KMO, Figure 3b, also show positive signal in the area of the cryptic pocket positive labels. However, the PM inference result shows significant positive signal in many other places on the protein. While some of these positive signals in the unknown region may reveal new cryptic pockets, there are so many that it is difficult to motivate any particular starting point for drug design programs. There may also be many false positives. Distillation of this result into actionable insights is therefore not straighforward as with ESP.

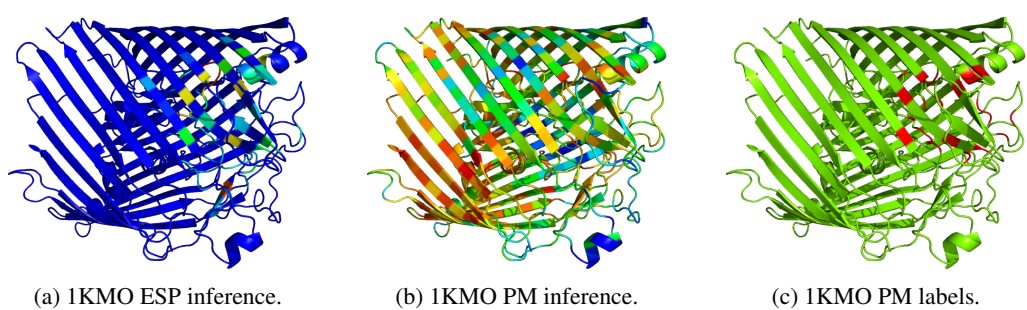

(a) 1KMO ESP inference.

(b) 1KMO PM inference.

(c) 1KMO PM labels.

Figure 3: Figure 3a shows ESP inference results, blue being negative prediction and red being positive. Figure 3b shows PM inference results using the same color scale. Figure 3c shows the binary PM test labels, where red indicates residues known to be associated with cryptic pocket formation, and green indicates unknown status and is masked during APS and AUC calculation.

### 3.3.2 BOVINE TRYPSIN (1BTP)

Figure 4 shows inference results and labels for bovine trypsin (RCSB PDB ID 1BTP). The ESP inference results for 1KMO, Figure 4a, show negative signal focued in the area of the non-pocket, negative labels as shown in Figure 4c, which are the PM test set labels for this protein. ESP predicts that no cryptic pockets are present in the area of the non-pocket, negative labels. This prediction is also easy to interpret, and phenomenologically correct in the sense that the known non-pocket areas stand out as such.

The PM inference results for 1BTP, Figure 4b, show positive signal broadly distributed across many residues with non-pocket negative labels. Whereas in the above example there were many positive predictions in an unknown area, here many positive predictions are in known non-pocket regions and are therefore clear false positives. Distillation of this PM inference result into actionable insights is therefore not possible because of false positives.

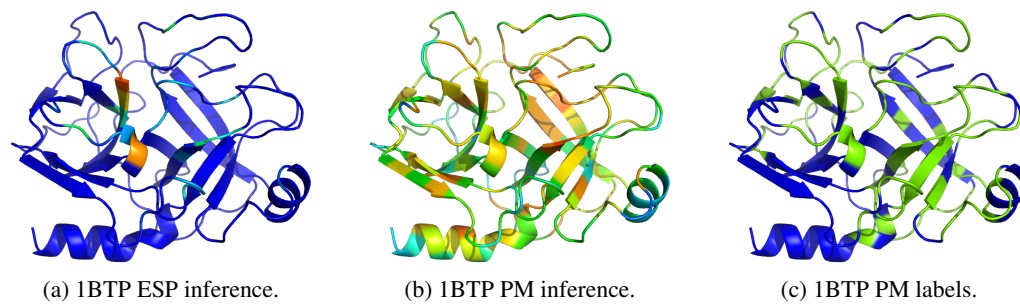

(a) 1BTP ESP inference.  (b) 1BTP PM inference.  (c) 1BTP PM labels.

Figure 4: Figure 4a shows ESP inference results (colors as above). Figure 4b shows PM inference results (colors as above). Figure 4c shows the binary PM test labels, where blue indicates residues known to not be associated with cryptic pocket formation, and green as above.

### 3.4 RESULTS SUMMARY

Using PDBBind-derived samples with less that 30% sequence identity with validation or test set samples, ESP with Ankh-Large achieves APS and AUC of 0.85 and 0.93, repecively. Using the same sequence identity thresold, ESP with ESM-2 15B achieves the best APS of all PLMs tested of 0.86, but at considerably higher computational cost due to the large PLM itself and embedding size of 5120, compared to Ankh's embedding size of 1536. MHA prediction heads tend to outperform others except for ProtT5-XL which favors MLP. Multi-task training using the ESM-2 SSP dataset produced the best model in many cases, but not the majority.

SOTA, PocketMiner, achieves 0.81 and 0.87 APS and AUC on its test set. The PM test set was also used for all ESP APS and AUC calculations. All PLMs except ProtBert enabled models that outperformed SOTA on one or both of APS and AUC. All Ankh-Large enabled models achieved the same. Figure 5 shows ROC and precision-recall (PR) curves for the best Ankh-Large models trained with 30% and 100% sequence identity thresholds and PocketMiner, as evaluated using the PocketMiner test set.

Inference via ESP tends to produce positive signal in a more focused and spatially-locallized manner, whereas inference via PM tends to produce positive signal broadly and smoothly distributed over many areas of a protein. While inference via ESP may miss some cryptic pockets, PM inference may not be favored for pharmaceutical discovery due to the quantity and distribution of positive predictions and non-trivial incidence of false positives.

## 4 CONCLUSION

We find that: (1) many PLMs enable ESP predictive efficacy beyond previous SOTA cryptic pocket prediction algorithms, and Ankh-Large and ESM-2 15B enable top AUC and APS, respectively, (2) that multi-task learning using the ESM-2 SSP dataset enhances predictive efficacy for many cases,

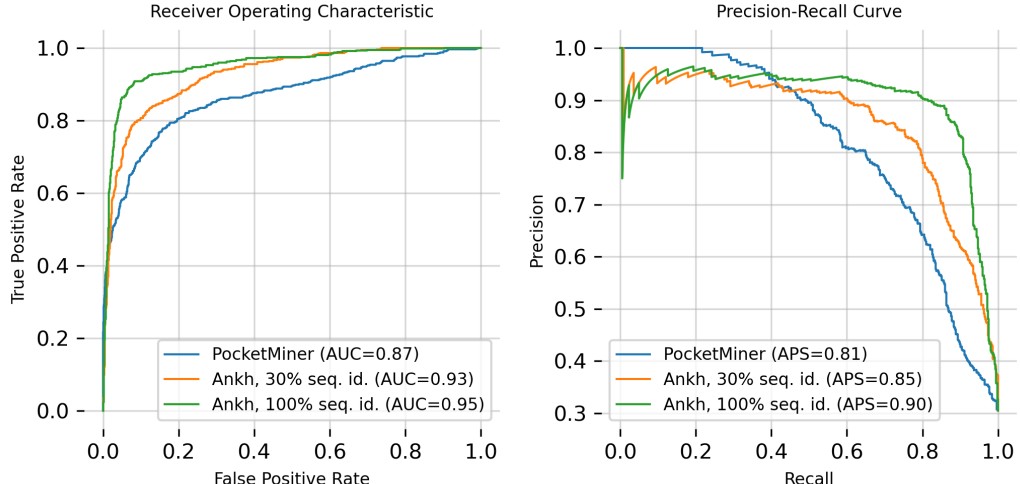

Figure 5: ROC and PR comparison between PocketMiner, ESP with Ankh-Large and 30% sequence identity threshold, ESP with Ankh-Large and 100% sequence identity threshold.

and (3) that ESP outperforms PM by a significant margin (ROCAUC 0.93 vs 0.87) on its own test set.

Because mere logistic regression also achieved a meaningful AUC, this result shows that Ankh-Large learned residue-level information from unsupervised training that linearly correlates with cryptic pocket formation propensity.

A small molecule pharmaceutical development program invests significant capital in exactly one target. Therefore prediction clarity and ease of interpretation are essential. Initiating drug development against a false positive cryptic pocket prediction would be very costly in terms of capital, time, and leadership bandwidth. False negative predictions are less costly in direct terms. Application of ESP at scale is therefore more valuable to small molecule drug developers than SOTA.

Future work can explore alternate labeling schema, including representations for multiple ligands for similar structures and using a residue's minimum distance to the nearest ligand atom to map labels into a continuous range. A PLM ensemble approach that combines embeddings from multiple PLMs and optionally downsamples may be worth exploring. It may also be useful to evaluate the efficacy of other PLMs in powering this application and explore potential for transfer learning between additional protein- and residue-level prediction tasks. Perhaps most significantly, future work may attempt to combine PLM- and MD-based approaches to achieve results outperforming either individual approach.

## 5 METHODS

### 5.1 DATASETS

Three datasets form the foundation for this work: (1) PDBBind (Su et al., 2019), (2) the ESM-2 SSP dataset, and (3) the PM validation and test sets. Training is conducted using the sequences and labels: (1) derived from the PDBBind dataset, and (2) directly from the ESM-2 SSP dataset. The PM validation and test sets serve those functions herein.

Since significant curation effort has been invested in the PDBBind dataset, we use it without further curation except for omission of proteins with synthetic residues. This results in a dataset of 17,986 complexes. Each protein's amino acid sequence is extracted from its structure file. The subject of missing residues is left to future work, rather we wish to test if PLMs can enable bypassing this step and still achieve useful results. We assign positive labels to any residue containing at least one atom within 6 Å of any ligand atom (Eguida & Rognan, 2022). We assign negative labels everywhere else.

The average sequence length is 292, total number of positive labels is 495,482, and total number of labels is 5,254,922.

The ESM-2 SSP dataset is used without modification, however since it is significantly larger than the PDBBind dataset we only use the 12,026 samples obtained using the "cv_partition=0" and "split=train" options.

The PM validation and testing data are used in the same way as by the PM authors in their work. Residues assigned an "unknown" or "unclassified" label are masked during loss calculation. This means that negative labels are only from rigid structures where the PM authors are confident no pocket can form, and positive labels are only associated with close proximity to ligands in resolved protein/ligand complexes. The validation and testing sets have 436 and 563 positive and 375 and 1,283 negative labels, respectively.

Data leakage is possible when identical sequences are present in the training set and either the validation or testing set. Because the PDB IDs are known for the PM validation and test sets, the authors used the RCSB APIs (Rose et al., 2021) to identify sets of structures in the PDBBind dataset that are within arbitrary seqeuence identity thresholds to the the PM validation and test structures. Sequence identity thresholds of 100%, 95%, 90%, 70%, 50%, and 30% were used. At training time, the structures within the desired identity threshold to the validation and test structures are removed from the training set. In addition to data leakage prevention, training using different sequence identity thresholds offers insight into generalizability of ESP and dependence of generalizability on the specific PLM used. The number of structures from the RCSB PDB database (Berman et al., 2000), PDBBind dataset, and PBDBind structures removed from the training dataset at different levels of sequence identity are reported in Table 6 (see Appendix).

## 5.2 ARCHITECTURE

Protein sequences extracted from PDBBind are input without modification into several PLMs: Ankh-Large, ESM-2 15B, ESM-2 3B, ProtT5-XL, and ProtBert. Fine-tuning is not executed; embeddings are calculated and stored, and then used to train a prediction head using the PDBBind dataset and optionally the ESM-2 SSP dataset. For ProtT5-XL, the average embedding is used as a pseudo-`[CLS]` token, as suggested by Ni et al. (2021).

PLM embeddings are then input into following prediction heads: Logistic regression (LR), multi-layer perceptrion (MLP) with one hidden-layer, a single layer of multi-headed-attention (MHA) not using the PLM's output `[CLS]` embeddings, and a single layer of multi-headed-attention (MHA) using the PLM's output `[CLS]` embeddings. The number of learnable parameters per prediction head for each PLM in the single-task setting is presented in Table 7 (see Appendix). The output of the prediction head is a residue-level cryptic pocket score in the single-task setting, and also SSP class likelihood in the multi-task setting.

## 5.3 TRAINING PROCESS

Training is executed using the PDBBind input data and labels derived via proximity to the ligand using binary cross entropy loss. When training concurrently with the SSP data, loss for SSP is evaluated via cross entropy loss. The total loss is a sum of the two, and a coefficient of SSP loss is used to adjust the relative significance of each loss. For results presented here, the SSP loss coefficient used is 1.0.

SGD has been used with no weight decay and a momentum value of 0.9.

Results for each specific PLM, prediction head, and task configuration are reported for the best model from 7 trials. Each trial is limited to a maximum of 40 epochs of training. We define a Figure of Merit (FOM) using the validation APS and AUC as shown in Eq 1. Several early stopping criteria have been implemented: detection of any NaN FOM, identical FOM for two consecutive epochs, or FOM increasing beyond 105% of an individual trial's lowest FOM. This strategy was chosen for convenience, since occurrance of any of these conditions tended not to lead to meaningful results.

$$\text{FOM} = 2 - \text{APS} - \text{AUC} \tag{1}$$

ACKNOWLEDGMENTS

The authors would like to thank Greg Bowman, Artur Meller, and the other authors of the Pock-etMiner manuscript, for their excellent work on the topic of cryptic pocket prediction and open-sourcing of their dataset and code. Protein structure visualizations are generated using PyMOL (Schrödinger, LLC, 2015).

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

Table 2: Average prediction score and ROCAUC results across architectures for ESP using ESM-2 15B embeddings. Training samples with greater than 30% sequence identity with any member of validation or testing sets has been removed.

| PLM: ESM-2 15B | PDBBind only | | w/ SSP | |
|---|---|---|---|---|
| Architecture | APS | AUC | APS | AUC |
| LR | 0.745 | 0.863 | 0.746 | 0.863 |
| MLP 16 | 0.784 | 0.875 | 0.780 | 0.871 |
| MLP 64 | 0.774 | 0.886 | 0.778 | 0.874 |
| MLP 256 | 0.810 | 0.897 | 0.800 | 0.894 |
| MLP 1024 | 0.830 | **0.914** | **0.835** | 0.908 |
| MHA 4 no CLS | 0.830 | 0.896 | 0.814 | 0.893 |
| MHA 8 no CLS | **0.865** | **0.911** | 0.817 | 0.885 |
| MHA 16 no CLS | 0.758 | 0.878 | 0.792 | 0.883 |
| MHA 4 | 0.816 | **0.897** | 0.801 | 0.883 |
| MHA 8 | **0.842** | 0.896 | 0.794 | 0.880 |
| MHA 16 | 0.773 | 0.874 | 0.795 | 0.885 |

Table 3: Average prediction score and ROCAUC results across architectures for ESP using ESM-2 embeddings. Training samples with greater than 30% sequence identity with any member of validation or testing sets has been removed.

| PLM: ESM-2 3B | PDBBind only | | w/ SSP | |
|---|---|---|---|---|
| Architecture | APS | AUC | APS | AUC |
| LR | 0.686 | 0.837 | 0.686 | 0.838 |
| MLP 16 | 0.754 | 0.857 | 0.758 | 0.859 |
| MLP 64 | 0.800 | 0.873 | 0.773 | 0.863 |
| MLP 256 | 0.797 | 0.877 | 0.782 | 0.882 |
| MLP 1024 | **0.820** | **0.892** | 0.783 | 0.866 |
| MHA 4 no CLS | **0.835** | **0.906** | 0.816 | 0.888 |
| MHA 8 no CLS | 0.806 | 0.886 | 0.785 | 0.877 |
| MHA 16 no CLS | 0.770 | 0.872 | 0.786 | 0.868 |
| MHA 4 | 0.799 | 0.883 | 0.791 | 0.882 |
| MHA 8 | **0.823** | **0.896** | 0.788 | 0.884 |
| MHA 16 | 0.749 | 0.858 | 0.778 | 0.879 |

# A  APPENDIX

Precision-Recall (PR), receiver operating characteristic (ROC), receiver operating characteristic area under-the-curve (AUC), and average precision score (APS) were calculated using Scikit Learn (Pedregosa et al., 2011).

Performance metrics for ESP using ESM2 with 15 billion parameters are summarized in Table 2.

Performance metrics for ESP using ESM2 with 3 billion parameters are summarized in Table 3.

Performance metrics for ESP using ProtT5-XL are summarized in Table 4

Performance metrics for ESP using ProtBert are summarized in Table 5.

For each pair of PLM and sequence identity limit prediction heads under test were: logistic regression, MLPs with widths of 16, 64, 256, and 1024, and MHAs with 4, 8, and 16 attention heads (for both the with and without `[CLS]` embedding cases).

Table 4: Average prediction score and ROCAUC results across architectures for ESP using Prot-Bert embeddings. Training samples with greater than 30% sequence identity with any member of validation or testing sets has been removed.

| PLM: ProtT5-XL | PDBBind only | | w/ SSP | |
|---|---|---|---|---|
| Architecture | APS | AUC | APS | AUC |
| LR | 0.803 | 0.884 | 0.800 | 0.885 |
| MLP 16 | 0.814 | 0.893 | 0.795 | 0.885 |
| MLP 64 | 0.813 | 0.889 | 0.827 | 0.897 |
| MLP 256 | 0.816 | 0.889 | 0.832 | **0.898** |
| MLP 1024 | 0.815 | 0.893 | **0.837** | 0.898 |
| MHA 4 no CLS | 0.801 | 0.887 | **0.822** | 0.894 |
| MHA 8 no CLS | 0.813 | 0.890 | 0.808 | **0.897** |
| MHA 16 no CLS | 0.803 | 0.889 | 0.813 | 0.892 |
| MHA 4 | 0.799 | 0.876 | 0.787 | 0.881 |
| MHA 8 | 0.780 | 0.892 | 0.778 | 0.887 |
| MHA 16 | **0.807** | 0.890 | 0.788 | **0.892** |

Table 5: Average prediction score and ROCAUC results across architectures for ESP using Prot-Bert embeddings. Training samples with greater than 30% sequence identity with any member of validation or testing sets has been removed.

| PLM: ProtBert | PDBBind only | | w/ SSP | |
|---|---|---|---|---|
| Architecture | APS | AUC | APS | AUC |
| LR | 0.576 | 0.725 | 0.576 | 0.733 |
| MLP 16 | 0.606 | 0.768 | 0.602 | 0.767 |
| MLP 64 | 0.653 | 0.809 | **0.676** | **0.815** |
| MLP 256 | 0.643 | 0.796 | 0.639 | 0.797 |
| MLP 1024 | 0.659 | 0.800 | 0.658 | 0.801 |
| MHA 4 no CLS | 0.639 | 0.791 | 0.613 | 0.764 |
| MHA 8 no CLS | 0.633 | 0.800 | 0.650 | 0.787 |
| MHA 16 no CLS | **0.689** | **0.820** | 0.669 | 0.800 |
| MHA 4 | 0.659 | 0.817 | 0.652 | 0.794 |
| MHA 8 | 0.661 | 0.813 | 0.654 | 0.790 |
| MHA 16 | **0.710** | **0.827** | 0.663 | 0.807 |

Table 6: Structures related to PM validation and test set structures at different thresholds of sequence identity.

| Sequence Identity % | RCSB PDB Structures | PDBBind Structures | PDBBind % removed |
|---|---|---|---|
| 100 | 4507 | 1104 | 6.14% |
| 95 | 6758 | 1894 | 10.5% |
| 90 | 6940 | 1928 | 10.7% |
| 70 | 8202 | 2009 | 11.2% |
| 50 | 9064 | 2294 | 12.8% |
| 30 | 22392 | 4937 | 27.4% |

Table 7: Trainable parameters per prediction head for different architectures and PLMs, without concurrent SSP training.

| Prediction Head | ProtBert/ProtT5-XL | Ankh | ESM-2 3B | ESM-2 15B |
|---|---|---|---|---|
| LR | 1,025 | 1,537 | 2,561 | 5,121 |
| MLP 16 | 16,417 | 24,609 | 40,993 | 81,953 |
| MLP 64 | 65,665 | 98,433 | 163,969 | 327,809 |
| MLP 256 | 262,657 | 393,729 | 655,873 | 1,311,233 |
| MLP 1024 | 1,050,625 | 1,574,913 | 2,623,489 | 5,244,929 |
| MHA | 4,197,376 | 9,441,792 | 26,222,080 | 104,872,960 |

