# OpenReview forum: "Protein Language Models Enable Accurate Cryptic Ligand Binding Pocket Prediction"
_ICLR.cc/2024/Conference — Submitted to ICLR 2024_

### Official Review · Reviewer_YdiM · 2023-10-17

**Soundness:** 1 poor
**Presentation:** 3 good
**Contribution:** 1 poor
**Rating:** 3
**Confidence:** 5

**Summary:**

Prediction of protein-ligand binding pocket is essential in protein functional analysis and small molecule pharmaceutical design. Previous methods rely on MD to discover cryptic pocket, which has poor scalability and bias. This paper proposes to use pretrained protein language models (PLMs) to tackle this problem and state a significant improvement in this challenging task over existing baselines.

**Strengths:**

(1) This article tries to address an important question, namely, identifying hard-to-find or cryptic protein pockets.

(2) The authors have considered the possibility of data leakage and made an attempt to remove similar structures in PDBBind from the training dataset.

(3) The experimental results are convincing and easy to understand for comparison.

**Weaknesses:**

(1) First and foremost, the novelty is quite limited. As a reviewer of ICLR, one of the top AI conferences, I expect to see great contributions to the deep learning community. However, the PLM algorithms including ESM-2, AnKh, ProtBert, and others are very mature techniques. The multi-task training with a secondary structure prediction is kind of interesting but still trivial. The author ought to provide more discussion and deeper insight regarding the benefit of this multi-task scheme in the main text. To summarize, the author just examine an existing methodology (i.e., PLMs) on some publicly available datasets (PDBBind + ESM-2 SSP) with a straightforward training paradigm (fixed embedding and [CLS] for prediction).

(2) It really confused me that whether the author aims to predict ligand-specific or ligand-free pockets. To be explicit, can different small molecules bind to different pockets in the same protein? If so, should we predict pocket based on the information of ligand rather than solely on the receptor? If not, that is, all molecules bind in the same position of the protein. The author still needs to union the positions to obtain the final pocket. However, the approach did not utilize any ligand representations.

(3) PLMs have shown promise in solving biological problems. However, it utilizes no structural information. Since we do have structural data in PDBBind, it really cannot convince me that PLMs are the best solution to predict cryptic pockets. (i) Please examine and compare more baselines (I am not familar with this specific task, and the references listed may be wrong) [A][B]. (ii) Machine-learning-based docking method can be also transferred to predict the pocket, such as EquiBind, and DiffDock. Please examine them. (iii) Please equip geometric networks (e.g., PocketMiner) with PLMs and verify the performance. See [C] for more details. I suppose PLMs + geometric models should outperform PLMs.

[A] Accelerating Cryptic Pocket Discovery Using AlphaFold.

[B] Graph Attention Site Prediction (GrASP): Identifying Druggable Binding Sites Using Graph Neural Networks with Attention

[C] Integration of pre-trained protein language models into geometric deep learning networks


(4) As a machine learning paper, it is very necessary to offer adequate experimental details for readers to reproduce the results (at least in the Appendix). For instance, how many computational resources (e.g., GPUs) have been used in the training? What is the optimizer type and how high is the learning rate?

(5) I believe it is better to clearly explain the types of PLMs that are used in the evaluation (at least in the Appendix). For instance, Ankh is new to me, and I have to search for details of this model myself. Besides, ESM-2 has no reference when the first time it appears.

(6) Make sure words are spelled correctly. For example, on Page 9 'PBDbind' should be 'PDBbind'.

**Questions:**

(1) In Section 5.2, the author uses the average embedding as a pseudo-[CLS] token for ProtT5-XL. However, in the official document of ESM-2, it is also recommended to use the average embedding as the representation of the entire protein. Have the author tried this style for training?

(2) A minor point is that the authors adopt an unusual arrangement to organize the sections, which accords with the template of Spring Nature Journals. From my humble point of view, a standard format of AI conference is to put the method section before the result and conclusion sections.

---

> ### Author Response · Authors · 2023-11-23
>
> Thank you for your time and consideration in reviewing our work.
>
> This is an application paper, and as such it is not the intent to showcase a new algorithm.  This is within scope as outlined in the call for papers.  BERT was not a new algorithm either, it was a new application, trained with publicly available data.
>
> Outperforming SOTA by such a large margin, on such an impactful application in small molecule pharmaceutical discovery is a noteworthy contribution to ML that should encourage many avenues of future work and be a net benefit for the human race.
>
> Regarding SSP, it was not consistently a benefit, however this is important to report for future investigators.
>
> The closest related work is PocketMiner, which has the goal of identifying residues--but not any ligand--that are likely to participate in or comprise cryptic pockets.
>
> We mentioned as future work the combination of this approach with geometric approaches.
>
> Equibind and Diffdock are fundamentally different approaches concerned with finding a binding pose given a protein and a ligand, which is a question that this work is not trying to answer.  If this were a docking application then they would be more relevant.
>
> Regarding implementation and computational details, please see the methods section for details on the optimizer.  The learning rate required some experimentation and was not the same for all model types presented.  The model sizes are described and models were trained on single GPUs (no DDP or similar necessary).
>
> References to all PLMs are included, and a detailed discussion of them would not leave room sufficient to explain the work done.
>
> Regarding specific questions:
> 1.  No ESM-2 embeddings were not averaged, although that could be added to the current results.
> 2.  The methods section can be rearranged to be after the background.

---

> ### Public Comment · ~Brian_Wiley1 · 2024-04-11
> **"To be explicit, can different small molecules bind to different pockets in the same protein?"**
>
> Anything can happen.  This is complicated stuff that needs to be explained in as simple of terms as possible to attract more people for making life saving medicine.
>
> Here is an example where imatinib with can inhibit BCR-Abl fusion in young children with leukemia (ALL) but if it binds in the allosteric site it can actual ACTIVATE it.  A complete reverse of biophysical phenomenon!
>
> Imatinib can act as an allosteric activator of Abl kinase
> https://www.ncbi.nlm.nih.gov/pmc/articles/PMC8752476/#:~:text=Imatinib%20is%20an%20ATP%2Dcompetitive,activator%20of%20the%20kinase%20activity.

---

### Official Review · Reviewer_1nRq · 2023-10-30

**Soundness:** 2 fair
**Presentation:** 2 fair
**Contribution:** 1 poor
**Rating:** 3
**Confidence:** 3

**Summary:**

This work presents a study of using pre-trained protein PLM models for cryptic binding pocket prediction tasks. The authors explored multiple pretrained models, fine-tuning strategies, architectures, and datasets. Some of the models achieved SOTA results.

**Strengths:**

- Cryptic ligand binding is an under-explored problem. This work could have important real word impact.
- The authors studied multiple settings and conducted many experiments.

**Weaknesses:**

From machine learning perspective, the technique contribution is limited. Using pretrained model on data-limited tasks is a well-studied approach. The combination of a PLM with MLP/MHA, or a SSP is not novel. The authors are encouraged to design some methods that are best for this specific task.

**Questions:**

- The color in Fig 3 is confusing. In caption it says, "blue being negative prediction and red being positive." But in picture lots of area are actually green.

- Is this the first work that use the concept of "cryptic pocket"? Seems no previous work about this term is referenced in Sec 2.1. The authors should either (a) discuss more previous study of cryptic pocket, or (b) give more accurate definition and explanation about this concept.

---

> ### Author Response · Authors · 2023-11-23
>
> Thank you for your thoughtful review.
>
> While the combination of PLM with MLP/MHA (with our without SSP) is not novel from an algorithms point of view, it is a unique approach that outperforms SOTA by a significant margin for an impactful application.
>
> While future work is envisioned to create bespoke algorithms, this work already constitutes a significant advance in this area.
>
> Regarding the figures, there are two types of figures.  The ones on the left and middle show predictions ranging from zero to one and represented by a colormap with blue as zero and red as one (with green as 0.5).  The rightmost of each group of three protein visualizations is the test set ground truth--which does have it's own particular definition.  The test set proteins can be grouped into two types of samples.  The first type has positive signal for residues labeled as definitively being associated with cryptic pockets, and in those samples green means "unknown," and that the green residues are masked such that no loss is incurred for any prediction for those residues.  Scoring is only done for the positive (red) residues in that case.  The negative examples are the opposite, where blue residues in negative examples are definitively associated with not being associated with cryptic pockets, and green is unknown and masked as before.
>
> Regarding cryptic pockets, section 2.1 is explicitly about them.  The references of PocketMiner and CryptoSite provide additional background, and are studies aimed at identifying them.

---

### Official Review · Reviewer_tBzC · 2023-10-30

**Soundness:** 2 fair
**Presentation:** 2 fair
**Contribution:** 2 fair
**Rating:** 3
**Confidence:** 3

**Summary:**

The text introduces a new method, ESP, that improves the prediction of protein-ligand binding pockets. It uses pretrained language models and the result is stronger than traditional SOTA method.

**Strengths:**

- It's interesting to see that PLMs have a stronger ability to predict cryptic binding sites than traditional SOTA methods.
- Sometimes, incorporating SSP for multitask learning can be beneficial for the task.

**Weaknesses:**

- The novelty appears limited. The main technique in this study is "PLM as feature," which emerged in 2018 (BERT).
- Additional ablation studies are necessary. Why not fine-tune the language model?
- Incorporating SSP doesn't seem consistently better than other methods.
- There's still room for improvement in the writing. There are too many technical details that aren't informative, such as comparative discussions on LR, MLP with MHA-x, with or without CLS. Also, the results across the tables lack consistency. The organization of the paper (i.e., "Results -> Conclusion -> Method") seems unusual to the ML audience.

**Questions:**

See weaknesses.

---

> ### Author Response · Authors · 2023-11-23
>
> Thank you for your review.
>
> While the combination of PLM with MLP/MHA (with our without SSP) is not novel from an algorithms point of view, it is a unique approach that outperforms SOTA by a significant margin for an impactful application.  BERT, as you mention, itself is not novel from an algorithms point of view either.
>
> What specifically would you ablate?
>
> Fine-tuning significantly escalates resource requirements, but is envisioned as future work given sufficient resources.
>
> SSP is still worth reporting to assist future investigators in their efforts.
>
> Happy to put methods after background.

---

### Official Review · Reviewer_3uqW · 2023-11-01

**Soundness:** 3 good
**Presentation:** 3 good
**Contribution:** 2 fair
**Rating:** 5
**Confidence:** 4

**Summary:**

This paper introduces Efficient Sequence-based cryptic Pocket prediction (ESP), which is a method for predicting cryptic binding pockets on protein sequences using pre-trained protein language models (PLMs). The embeddings drawn from the PLM are fed into a prediction head, which is responsible for predicting the presence of a cryptic pocket at each residue. Optionally, a multi-task setting is also considered, where the secondary structure of the protein is also predicted using the PLM embeddings. Several popular PLMs and prediction heads are evaluated, and it is shown that with the right combination of PLM and prediction head, the proposed method outperforms the state-of-the-art PocketMiner baseline in cryptic pocket prediction performance.

**Strengths:**

The main strength of the paper is the significant performance boost over the SOTA PocketMiner approach. As the authors allude to, protein language models have shown great success in computational studies of protein structure and function, and this work is another example where pre-trained PLMs shine. The proposed method could potentially impact the design of drugs that could bind to proteins through cryptic ligandable pockets.

**Weaknesses:**

I believe the main weakness of the paper is that, in terms of the proposed method, the paper does not provide a significant novel contribution to the broader ICLR community. This is understandable since the purpose of the paper is to showcase that PLMs could be beneficial in identifying cryptic pockets in proteins, but I wonder whether ICLR is the best venue for such work to be published and gain visibility.

**Questions:**

- The authors mention that ESP provides more localized positive cryptic pocket predictions as compared to PocketMiner, whose positive predictions are more broadly distributed. Given the knowledge that cryptic pockets tend to have local structures, could such prior knowledge be injected into the prediction model, for example, as a regularizer (where local positive predictions are encouraged during model training, while negative ones are discouraged)?

- Could you please provide more details on the secondary structure prediction (SSP) task? How many classes is this task composed of? Is there a separate prediction head for structure prediction on top of the cryptic pocket prediction head? Why and how is its weight in the objective function chosen to be 1?

- In Section 5.3, it is mentioned that the results are reported for the *best* model from 7 trials. I was expecting that for each specific configuration (i.e., PLM/head/task), the *average* performance (and its *standard deviation*) across the seven random trials would get reported, not the *maximum* performance.

- The number of parameters for ESP in Table 7, especially with the MLP 1024 and MHA heads, is orders of magnitude larger than the number of parameters of PocketMiner. This is not even taking into account the number of PLM parameters (which are taken to be frozen in this paper). How does ESP perform compared to PocketMiner for a comparable number of parameters?

- I may have missed these, but there are certain acronyms in the manuscript that are not defined anywhere in the paper (such as AUC, APS, LR, CLS). Please review the manuscript and make sure all acronyms are defined the first time they are used.

---

> ### Author Response · Authors · 2023-11-23
>
> Thank you for your effort in reading and considering our work and its impact.
>
> We appreciate your concern regarding algorithmic novelty and venue.  Our thinking is that if something simple significantly outperforms SOTA for an impactful application then that's an important contribution.  It also paves the way for future work to build upon the lessons learned, one of which being the marginal benefit of SSP in this context.
>
> That's definitely an interesting idea, to regularize based on the overall size and number of residues comprising ground truth cryptic pockets.  This work focused on using the simplest possible process, and would be a relevant baseline for introduction of a regularizer as you describe.
>
> SSP is a relatively basic and low impact task whereby each residue is a member of eight or nine classes (depending on formulation).  ESM-2 SSP classes start with the 8 canonical DSSP classes:
>     G = 3-turn helix (310 helix). Min length 3 residues.
>     H = 4-turn helix (α helix). Minimum length 4 residues.
>     I = 5-turn helix (π helix). Minimum length 5 residues.
>     T = hydrogen bonded turn (3, 4 or 5 turn)
>     E = extended strand in parallel and/or anti-parallel β-sheet conformation. Min length 2 residues.
>     B = residue in isolated β-bridge (single pair β-sheet hydrogen bond formation)
>     S = bend (the only non-hydrogen-bond based assignment).
>     C = coil (residues which are not in any of the above conformations)
> (ref:  https://en.wikipedia.org/wiki/Protein_secondary_structure)
>    And the ESM-2 SSP dataset also uses "X" as unknown, making for 9 classes overall.
>  The SSP class outputs are in a multi-task setup with the cryptic pocket prediction, sharing everything except the final output neuron.  It's relative weight in the objective function was experimented with and empirically determined to produce best results at 1.
>
> While average and standard deviation are commonly used in ML literature, because this is a small molecule pharmaceutical application, the performance of the best model is the actual limiting factor in real-world application.  Evaluating methodologies by their best model is therefore most informative regarding practice.
>
> It's definitely a valid point regarding the vast expansion of the trainable parameter space compared to PocketMiner.  However, for Ankh, even logistic regression makes an improvement to ROCAUC.  This does not take into account the parameters of the PLM.  However, it's not clear that PocketMiner can meaningfully scale to larger numbers of trainable parameters in the sense that graph methods have scaling issues regarding run time, and more parameters doesn't always mean more efficacy.  This is an interesting discussion to include in more detail in future work.
>
> We are happy to explicitly define those acronyms, either in the main text or an appendix.
>
> Thank you again for your thoughtful feedback, and looking forward to your reply.

---

### Author Response · Authors · 2023-11-23

Thank you all for your time and consideration in reviewing our work.

We understand the concern regarding algorithm innovation or the lack thereof in this work.  BERT was not an innovative algorithm in the sense that it is just Transformer encoder blocks, the Cloze task is very old, and there are many works with arbitrary training sequences and prediction heads.  This is an application paper, which according to our reading of the call for papers, is within scope for ICLR 2024.

Our view is that because the method described significantly outperforms SOTA that it is a meaningful contribution to the ML community.  If it is simple and does so, then all the better.  That is even more important for investigators to know.

At the end of the day, this work significantly outperforms SOTA in a critical application that aids development of small molecule pharmaceutical drugs to help patients fighting cancer, immunological, neurodegenerative, and other diseases.  It provides a new and better baseline for future work in this space.

---

> ### Public Comment · ~Brian_Wiley1 · 2024-04-11
> **Agree**
>
> The application far outweighs the theory.  We are trying to get personalized medicine into trials to save lives.  Not prove novelty.  A publication of extreme theoretical novelty doesn't help the person who is too weak to take chemo because they regimen is too toxic.  As Nobel Prize winner Dr. William Kaelin said "It takes more guts and more courage to accept a paper than it does to reject it."

---

### Meta-Review · Area_Chair_QVL9 · 2023-12-11

**Metareview:**

All three reviewers raised concerns about the novelty of the approach, stating that the paper does not provide a significant novel contribution to the broader ICLR community, and the technique contribution from a machine learning perspective is limited. Reviewer 2 also pointed out the need for additional ablation studies and improvements in the writing.

**Justification For Why Not Higher Score:**

This work is a direction application of large LMs in physical sciences.

We acknowledge that the architecture of BERT, which employs Transformer encoder blocks and the Cloze task, does not in itself constitute an innovative algorithm. However, the groundbreaking aspect of BERT lies in its unique demonstration of the effectiveness of a masked language model, marking a first in the ML and NLP field; such a huge success of masked LM has not been observed in any other fields.

**Justification For Why Not Lower Score:**

N/A

---

### Decision · Program_Chairs · 2024-01-16

Reject